- **1** Technical Note: Rapid Normal-phase Separation of Phytoplankton
- Lipids by Ultra-High Performance Supercritical Fluid
- Chromatography (UHPSFC)
- 5 J. Brandsma<sup>1</sup>, T. R. Sutton<sup>2,a</sup>, J. M. Herniman<sup>3</sup>, J. E. Hunter<sup>3,a</sup>, T. E. G. Biggs<sup>4</sup>, C. Evans<sup>4,b</sup>, C. P.
- D. Brussaard<sup>4,5</sup>, A. D. Postle<sup>1</sup>, T. J. Jenkins<sup>6</sup>, and G. J. Langley<sup>3</sup>
- <sup>1</sup> Clinical and Experimental Sciences, Faculty of Medicine, University of Southampton, Southampton,
- United Kingdom
- <sup>2</sup> NIHR Southampton Biomedical Research Centre, University of Southampton and University
- Hospital Southampton NHS Foundation Trust, Southampton, United Kingdom
- <sup>3</sup> Chemistry, Faculty of Natural and Environmental Sciences, University of Southampton,
- Southampton, United Kingdom
- <sup>4</sup> NIOZ Royal Netherlands Institute for Sea Research, Texel, The Netherlands
- <sup>5</sup> Aquatic Microbiology, Institute for Biodiversity and Ecosystem Dynamics, University of Amsterdam,
- Amsterdam, The Netherlands
- <sup>6</sup> Waters Corporation, Wilmslow, United Kingdom
- <sup>a</sup> current address: Clinical and Experimental Sciences, Faculty of Medicine, University of
- Southampton, Southampton, United Kingdom
- <sup>b</sup> current address: Natural Environment Research Council, National Oceanography Centre,
- Southampton, United Kingdom

Correspondence to: J. Brandsma (J.Brandsma@soton.ac.uk)

- Keywords: marine phytoplankton, lipids, supercritical fluid chromatography, mass spectrometry
- Short title: Phytoplankton lipid separation by UHPSFC

## 27 Abstract

- Lipid metabolism is one of the cornerstones of biochemistry, and these chemically diverse
- biomolecules play key roles in molecular physiology and mediate interactions between microbes and
- their environment that play out on cellular to ecosystem scales. Marine phytoplankton fix in the
- range of 1 billion tonnes of carbon as lipid biomass each year, which goes on to fuel higher trophic
- levels or ends up in the marine dissolved organic matter pool. Yet despite the importance of the vast
- marine lipidome for global biogeochemistry, surprisingly little is known about its diverse inventory of
- molecular structures, or the influence that dynamic environmental conditions exert on microbial
- lipid synthesis, remodelling and turnover.
- To aid in this research, a high-throughput platform for comprehensive analysis of phytoplankton
- lipids was developed using Ultra-High Performance Supercritical Fluid Chromatography (UHPSFC).
- This recently developed technology combines a primary supercritical fluid (CO<sub>2</sub>) mobile phase with
- an organic co-solvent of choice. Using a simple 10 minute gradient and a sub-2 µm particle column,
- UHPSFC efficiently separates all of the major neutral and polar lipid classes encountered in
- phytoplankton in a single analysis. These can then be measured by tandem mass spectrometry using
- established precursor and neutral mass loss scans.
- To demonstrate the analytical power of this novel platform the lipid compositions of a diverse range
- of phytoplankton species grown in culture, as well as phytoplankton community samples from the
- Western Antarctic Peninsula, were analysed. With higher chromatographic resolution and a much
- shorter analysis time than current liquid chromatography methods, the application of UHPSFC has
- considerable potential to benefit large-scale lipidomic studies, including in the field of environmental48 microbiology.

### 49 1 Introduction

- The field of lipidomics has advanced dramatically over the last two decades, spurred on by the 51 development of ever more sophisticated analytical instrumentation, in particular in the area of mass 52 spectrometry (extensively reviewed in Blanksby and Mitchell, 2010; Harkewicz and Dennis, 2011; 53 Köfeler et al., 2012). Although in some cases the direct analysis of crude lipid extracts by MS is 54 sufficient ('shotgun' lipidomics; Han et al., 2012), more in-depth analyses of highly complex lipid 55 mixtures will often require some form of chromatographic separation first. Gas chromatography 56 (GC) is widely used for separation of less polar, volatile compounds, such as fatty acid methyl esters 57 (FAMEs). In contrast, liquid chromatography (LC) is capable of separating more polar, non-volatile 58 compounds, and its compatibility with soft ionisation techniques such as electrospray ionisation (ESI) 59 or atmospheric pressure chemical ionisation (APCI), makes it the method of choice for most 60 lipidomics applications (Cajka and Fiehn, 2014). 61 An alternative to GC and LC is supercritical fluid chromatography (SFC), which uses CO<sub>2</sub> as the main 62 mobile phase. The CO<sub>2</sub> is heated and compressed beyond its supercritical point (31.1°C and 7.39 63 MPa), giving it a higher diffusion coefficient and lower viscosity than regular liquids. Due to its low 64 polarity (comparable to hexane) CO<sub>2</sub>-based SFC is an effective and rapid way of separating non-polar 65 analytes. Furthermore, the polarity range can be greatly increased by adding an organic co-solvent 66 such as methanol, and operating the system in a subcritical state. Sub-2 µm particle SFC columns 67 with improved separation performance have recently been introduced (Nováková et al., 2014). 68 These systems are known as Ultra-High Performance SFC (UHPSFC) or go by their commercial name 69 of Ultra-Performance Convergence Chromatography® (Waters ACQUITY UPC<sup>2</sup>) and represent a rapid 70 and versatile separation technique with many potential applications in analytical chemistry (Lesellier 71 and West, 2015; Nováková et al., 2014). The use of UHPSFC-MS systems for analysis of a range of 72 lipid classes has already been demonstrated (Bamba et al., 2008; Lee et al., 2012; Ratsameepakai et 73 al., 2015; Uchikata et al., 2012; Yamada et al., 2013; Zhou et al., 2014). However these applications 74 do not cover the main lipid classes that are synthesised by marine microbes and indeed predominate 75 throughout the marine environment. 76 Marine phytoplankton (photosynthesising unicellular microorganisms) are responsible for half the 77 global net primary production, fixing approximately 39 gigatonnes of carbon each year and 78 converting it into biomass (Falkowski et al., 2003; Field et al., 1998; Rousseaux and Gregg, 2014). 79 Approximately 10-20% of this phytoplankton biomass consists of lipids (Hedges et al., 2002), yet
- 80 surprisingly little is known at present about the composition, dynamics and biogeochemical roles of
- 81 the vast marine lipidome. Microbial lipid metabolism can be influenced by a wide range of factors,

3

including growth stage (Schwenk et al., 2013), dynamic environmental conditions (Gašparović et al., 2014; Guschina and Harwood, 2009; Van Mooy et al., 2009), or mortality processes (Evans et al., 83 84 2009; Hunter et al., 2015; Vardi et al., 2009). Furthermore, lipids and other metabolites are known to play important roles in cellular signalling and interactions (e.g., Fernandis and Wenk, 2007). 85 However, these have mostly been studied in the context of human biology and lipid signalling in 86 87 microbial ecosystem functioning has hardly been explored. Most of the carbon and nutrients fixed in the primary producers' biomass either sustains higher trophic levels within the marine food web, or 88 89 is recycled within the microbial loop (Mojica et al., 2015; Suttle, 2007). However, an estimated 15-20% of the fixed carbon persists in the oceans as refractory dissolved organic matter (RDOM) and 90 91 thereby forms a major sink for atmospheric  $CO_2$  (Hansell, 2012). Lipids comprise a significant part of 92 marine particulate and dissolved organic matter (Bianchi and Canuel, 2011; Lee et al., 2004), although their role in the RDOM pool is still unclear. Finally, through vertical transport and 93 94 sedimentation lipids end up in the geological records, where they provide a unique window on past 95 marine microbial communities and environmental conditions (Peters et al., 2007 and references 96 therein). 97 In this technical note UHPSFC is used to rapidly and efficiently separate all the major lipid classes

encountered in phytoplankton, prior to detection by mass spectrometry. The UHPSFC-ESI-MS/MS
platform described here has higher chromatographic resolution and a shorter analysis time than
current standard LC methods for phytoplankton lipid separation (e.g., Anesi and Guella, 2015;
Popendorf et al., 2013; Sturt et al., 2004) and adds versatility in the types of analytes that can be
separated. This technology enables higher sample throughput and provides increased analytical
flexibility, benefitting large-scale and/or in-depth lipidomics studies. It therefore has considerable
and widespread applications in the fields of environmental microbiology and biogeochemistry.

# 106 2 Materials and methods

# 107 2.1 Lipid standards, algal cultures and environmental samples

- The following synthetic phospholipid standards were obtained from Avanti Polar Lipids (Alabaster,
- AL, USA): 1,2-didodecanoyl-sn-glycero-3-phosphocholine (PC 12:0/12:0), 1,2-ditetradecanoyl-sn-
- glycero-3-phosphocholine (PC 14:0/14:0), 1-hexadecanoyl-2-eicosatetraenoyl-sn-glycero-3-
- phosphocholine (PC 16:0/20:4), 1,2-ditetradecanoyl-sn-glycero-3-phosphoglycerol (PG 14:0/14:0),
- 1,2-ditetradecanoyl-sn-glycero-3-phosphoethanolamine (PE 14:0/14:0), 1,2-ditetradecanoyl-sn-
- glycero-3-phosphoserine (PS 14:0/14:0), 1,2-dihexadecanoyl-sn-glycero-3-phosphoinositol (PI

- 16:0/16:0), and 1,2-ditetradecanoyl-*sn*-glycero-3-phosphate (PA 14:0/14:0). A single sphingomyelin
- (hexadecanoyl-sphingenine-phosphocholine SM d18:1/16:0) and diacylglycerol standard (1-
- stearoyl-2-arachidonoyl-sn-glycerol DAG 18:0/20:4) were also obtained from Avanti Polar Lipids.
- Three galactolipid standards, purified from spinach leaf extract, were obtained from Lipid Products
- (Redhill, Surrey, UK). They included the classes mono- and di-galactosyldiacylglycerol (MGDG and
- DGDG), and sulfoquinovosyldiacylglycerol (SQDG) with a range of fatty acids (predominately C34:3).
- A purified standard of the betaine lipid diacylglyceryl-carboxyhydroxymethylcholine (DGCC) was
- kindly donated by Benjamin Van Mooy (Woods Hole Oceanographic Institute, MA, USA; see
- Popendorf et al., 2013 for details). Again, this standard contained a range of fatty acids (from C30:3
- to C42:11).
- Cultures of seven ecologically relevant and/or model phytoplankton species were obtained from the
- Culture Collection of Algae and Protozoa (CCAP, Oban, UK): Chaetoceros calcitrans (CCAP1010/11),
- Ditylum brightwellii (CCAP1022/1), Micromonas pusilla (CCAP 1965/4), Phaeodactylum tricornutum
- (CCAP1055/1), Thalassiosira pseudonana (1085/12) and Thalassiosira weissflogii (CCAP1085/18). The
- remaining cultures of Emiliania huxleyi (RCC1228), Dunaliella tertiolecta (CCMP364), Synechocystis
- sp. (PCC6803) and Trichodesmium erythraeum (IMS101) were kindly provided by colleagues at the
- University of Southampton (see Acknowledgements).
- The five marine diatoms C. calcitrans, D. brightwellii, P. tricornutum, T. pseudonana and T. weissflogii 132 were batch cultured in f/2+Si media (Guillard, 1975) and incubated at 18°C, under a 12:12 light:dark 133 cycle with illumination of 123 µmol quanta m<sup>-2</sup> s<sup>-1</sup>. The Prasinophyte *M. pusilla* and Prymnesiophyte E. huxleyi were cultured under the same conditions as the marine diatoms, but in f/2-Si media 134 135 (Guillard, 1975). The Chlorophyte D. tertiolecta was grown in f/2-Si Media at 20-23°C, under a 12:12 136 light:dark cycle with illumination of 100 µmol quanta m<sup>-2</sup> s<sup>-1</sup>. The cyanobacterium Synechocystis sp. was grown in BG11 media (Stanier et al., 1971) with 5 mM glucose and buffered with 10 mM TES-137 138 KOH at pH 8.2. This culture was incubated at 30°C under constant illumination of 50  $\mu$ mol quanta m<sup>-2</sup> 139 s<sup>-1</sup> with shaking of 150 rpm. Finally, the filamentous cyanobacterium *T. erythraeum* was grown in 140 modified YBC-II media (Chen et al., 1996), under a 12:12 light:dark cycle with illumination at 130 141 µmol quanta m<sup>-2</sup> s<sup>-1</sup> at 27°C with shaking of 150 rpm. Cells from all phytoplankton cultures (total 142 volume varying from 20 to 50 ml per culture) were harvested during late exponential or early 143 stationary growth phase onto pre-combusted GF/F filters (47 mm diameter, 0.7 μm mesh size; Fisher 144 Scientific, Loughborough, UK; pre-combusted for 4h at 450°C), with the exception of T. erythraeum
- which was harvested by centrifugation at an unknown point of the growth phase.

- In addition to the cultures, three natural marine phytoplankton samples were obtained during the
- austral summer of 2012/2013 as part of the Rothera Oceanographic and Biological Time Series
- (RaTS), which is located in Ryder Bay on the Western Antarctic Peninsula (Clarke et al., 2008;
- Venables et al., 2013). Discrete sampling was performed at a depth of 15 m using a Niskin bottle
- deployed from a rigid inflatable boat. Particulate matter for lipid analysis was obtained by filtration
- of 2-5 I water samples over pre-combusted GF/F filters (47 mm diameter, 0.7 μm mesh size),
- followed by storage at -80°C for transport back to the UK. Although standard GF/F filters were used
- it is expected that a substantial portion of the sub-0.7 μm fraction of the microbial community (i.e.,
- prokaryotes) were retained as well due to the fairly large volume of water filtered.

#### 155 2.2 Sample extraction and preparation

- Phytoplankton cell pellets and the particulate matter filters were extracted using the standard Bligh-
- Dyer method (Bligh and Dyer, 1959) with a few minor modifications: (1) all monophasic extracts
- were centrifuged for 10 minutes at 3000 rpm to remove interfering particulates, and (2) the filtered
- RaTS samples were subjected to 30 sec ultrasonication after the first solvent addition step in order
- $\,$  to improve recovery of the lipids. Lipid extracts were dried under a stream of heated  $N_2$  gas and
- stored at -80°C. Prior to analysis the samples were reconstituted in 50 µl of MeOH (HPLC grade).

#### 162 2.3 Analytical platform and settings

- All method development and sample analysis was undertaken using a Waters ACQUITY UPC<sup>2</sup>
- interfaced with a Waters Xevo TQD tandem quadrupole mass spectrometer equipped with an ESI
- probe. Separation was achieved over a UHPSFC-specific column packed with a diol-functionalized
- bridged ethylene hybrid (BEH) particle (Waters Torus diol column; 3 x 100 mm, 1.7 μm particle size,
- 130 Å pore size) which was held at 40°C. The primary mobile phase (A) was supercritical  $CO_2$  (food
- grade) and the co-solvent (B) consisted of MeOH:H<sub>2</sub>O (98:2 v/v) with 50 mM ammonium acetate.
- The addition of a small amount of water markedly improved peak shapes whilst having only minor
- effect on the retention times. A make-up solvent of MeOH with 1% formic acid at a flow rate of 0.45
- 171 ml min<sup>-1</sup> was added prior to MS detection. The column was equilibrated for 2 min at 2% B prior to
- each analysis, while the analytical run itself comprised a 10 min linear gradient of 2 to 40% B at a
- flow rate of 1.5 ml min<sup>-1</sup>. The injection volume was 2  $\mu$ l and larger injections volumes were noted to
- be detrimental to overall peak shape.
- Retention times for the 11 different lipid classes listed above were optimised using the synthetic or
- purified lipid standards. In addition, retention times of the betaine lipids diacylglyceryl-
- trimethylhomoserine (DGTS) and/or diacylglyceryl-hydroxymethyl-trimethyl- $\beta$ -alanine (DGTA) were

- experimentally determined using the extracts of *M. pusilla*. The head groups of these lipid classes
- are structural isomers, and although DGTA is normally more abundant than DGTS in this particular
- algae (Maat et al., 2015) we did not conclusively achieve chromatographic resolution of these two
- classes without purified/synthetic standards. The neutral lipid (NL) classes di- and triacylglycerol
- (DAG and TAG), as well as sterol lipids, have been shown to elute rapidly at low co-solvent
- conditions (<10%; Zhou et al., 2014). This was confirmed on our system using the DAG standard and
- a total lipid extract of human blood plasma. Although not a focus of this study, it was noted that
- various classes of lysophospholipids elute between retention times of 6.5 and 8.5 minutes (data notshown).
- After separation the different lipid classes can be analysed by positive ion and negative ion
- electrospray tandem mass spectrometry. A series of specific scan events was set up to coincide with
- the different elution times (Table 1). Precursor and neutral mass loss scans were based on well-
- established fragmentation protocols (e.g., Brügger et al., 1997; Sturt et al., 2004; Popendorf et al.,
- 2013), although the neutral lipids (DAG and TAG) were only measured as [M+NH<sub>4</sub>]<sup>+</sup> ions in the full
- scan positive ion ESI MS due to poor fragmentation spectra.

#### 194 3 Results and discussion

## 195 3.1 Chromatography

UHPSFC is a normal-phase system that separates lipids primarily by the polarity of their head group 197 (i.e., lipid class), and to a lesser extent by their fatty radyl configuration (i.e., bond type, carbon chain 198 length, number of double bonds or side chains, etc.). A further useful attribute of UHPSFC is the 199 ability to influence component separation and elution by controlling the amount of organic co-200 solvent present in the mobile phase. This allows for the separation of apolar lipid classes at low co-201 solvent conditions (

- While this fraction can be separated in much greater detail using a shallow gradient of very low co-
- solvent concentrations (e.g., Zhou et al., 2014), in many cases full scan positive ion ESI data will yield
- sufficient compositional information.
- The polar lipid classes elute during the subcritical second part of the analysis, at co-solvent
- concentrations above 10-15%. The elution order is DGTS/DGTA > MGDG > PC > PG / DGCC / SM > PE
- 215 / PA > DGDG > SQDG > PI (Fig 1; Table 1). As no synthetic or purified standards were available for
- either DGTS or DGTA, the elution order of these isomeric betaine lipid classes could not be
- confirmed. However, as it is known how they separate in normal-phase LC systems (Dembitsky,
- 1996; Popendorf et al., 2013) it is likely that they behave similarly in SFC-based systems. Elution
- times are typically fast, with peak widths of single compounds between 2-4 seconds (full width at
- half maximum). As mentioned previously, the fatty radyl configurations of the different lipid species
- within a class has a small effect on their retention times. Increasing the number of double bonds in
- the fatty radyl chains increases the retention time (by about 4-5 seconds per double bond), whereas
- increasing the number of carbon atoms reduces it (by about 3 seconds per C<sub>2</sub>H<sub>4</sub> unit). The effects of
- other functional groups are unknown at this point. Most biological samples will show a degree of
- structural complexity within each lipid class, with a range of different lipid species being present.
- Using the UHPSFC platform described here, the elution time of a mixed composition lipid class is
- typically between 20 and 30 seconds, depending on complexity and sample concentration (Fig. 1). In
- contrast, on normal-phase LC systems this is in the range of 1-3 minutes. Peak shapes were generally
- sharp and symmetrical, although the PS and galactolipid standards (MGDG, DGDG and SQDG) had
- comparatively poor peak shapes with significant tailing. The same is observed in diol column-based
- LC systems, and these lipid classes would potentially benefit from being separated on a different
- column stationary phase.

## 233 **3.2** Application to phytoplankton cultures and community samples

- To test the UHPSFC-ESI-MS/MS platform the lipid compositions of a diverse range of phytoplankton
- species grown in culture were measured (Table 2). These analyses were performed for
- demonstrative purposes only and the data presented here is not an exhaustive characterisation of
- the lipidomes of these organisms. Although phytoplankton lipidomes differ significantly between
- species and are further dependant on a range of environmental factors (e.g., Van Mooy et al., 2009),
- they generally comprise a dozen or so neutral and polar lipid classes (Guschina and Harwood, 2009).
- These same classes have been shown to predominate throughout the marine environment
- (Brandsma et al., 2012; Popendorf et al., 2011; Schubotz et al., 2009; Van Mooy and Fredricks, 2010;

Wakeham et al., 2012), and were therefore specifically targeted in this study (e.g., PA, PS and PI

- were not measured).
- PC was the most common lipid class and present in high amounts in all phytoplankton species except 245 for Synechocystis sp. Instead, this cyanobacterium had PG as the main glycerophospholipid, which 246 was only detected in smaller amounts in the other phytoplankton species. PE was only found in 247 three species: T. weissflogii, M. pusilla and D. tertiolecta, whereas PA, PI and SM were not detected 248 at all. These results are generally in line with existing data on phytoplankton lipidomics (e.g., 249 Guschina and Harwood, 2009), with PCs being the predominant membrane lipids in the eukaryotes 250 but much less common in prokaryotes (Sohlenkamp et al., 2003). Glycerophospholipid fatty acid 251 compositions were inferred from the molecular masses, and comprised a wide range of carbon chain 252 lengths and number of double bonds, with combinations of C14:0, C16:0-C16:1, C18:0-C18:4, C20:5 253 and C22:6 fatty acids being the most common. 254 Betaine lipids were detected in all of the eukaryotic algae, although they were absent in the 255 cyanobacteria. DGTA and/or DGTS were particularly prominent in *M. pusilla*, but could not be 256 detected in the Thalassiosira diatoms. M. pusilla is reported to predominately synthesize DGTA
- (Maat et al., 2015), but as detailed in Sect. 2.3 we could not conclusively demonstrate which was
- present in each of the phytoplankton cultures. DGCC was only found in *T. pseudonana* and *E. huxleyi*,
- the latter of which had roughly equal concentrations of DGCC and PC. Furthermore, the fatty acid
- composition of DGCC was very similar to that of PC, which supports the idea that these two lipid
- classes substitute for each other in phytoplankton membranes (Hunter, 2015; Martin et al., 2011;
- Van Mooy et al., 2009).

The results for the main chloroplast galactolipids were somewhat mixed. In most of the phytoplankton species only small to medium amounts of MGDG and SQDG and trace amounts of DGDG could be detected. However, the cyanobacterium *Synechocystis sp.* contained large amounts of all three galactolipid classes, which is consistent with the comparatively high proportion of thylakoid membrane in cyanobacterial cells (Herrero and Flores, 2008). The fatty acid configurations of the galactolipids were overall more saturated than for the glycerophospholipids, with C14:0, C16:0-C16:1 and C18:0-C18:1 dominating, but comparatively low levels of the characteristic C18:3 (Kenyon, 1972).

- Finally, modest amounts of DAGs and TAGs were detected in almost all of the phytoplankton species.
- No fragmentation scans for full structural elucidation of the neutral lipids were performed at this time,
- but the fatty acids in these classes were comparable to those found in the polar lipid classes.

The marine microbial community samples from the RaTS site (Western Antarctic Peninsula) showed 274 a more complex lipid profile than any of the phytoplankton cultures (Table 2). The PC class was the 275 most diverse and ranged from PC 26:0 to PC 44:12, including a significant number of species with 276 short (e.g., C12:0, C13:0, C14:0-C14:1) and odd-chain fatty acids. PCs with longer chain and/or 277 polyunsaturated fatty acids (for example the observed PC 38:6, PC 40:10 and PC 44:12) are 278 characteristic of eukaryotic phytoplankton, such as the diatoms that dominate in carbon biomass at 279 this site (Clarke et al., 2008; Piquet et al., 2011). However, short and/or odd-chain fatty acids are 280 commonly associated with prokaryotic rather than eukaryotic membrane lipids. Although PCs are 281 generally rare in bacteria (Sohlenkamp et al., 2003), their presence thus suggests that prokaryotes 282 constitute a substantial proportion of the microbial community at the RaTS site, an observation 283 further supported by the presence of PG and DGTA/S with similar fatty acid compositions. 284 Cyanobacteria are typically poorly represented in polar marine waters (Vincent, 2000), but a variety 285 of other bacterial clades have been reported from this site (Piquet et al., 2011), and these were likely 286 retained on the GF/F filters that were used to collect the water samples. Of the remaining lipid 287 classes only trace amounts of PE, DGCC, SQDG and DGDG were detected, but MGDG was more 288 abundant, containing combinations of C16:0 and C18:0-C18:1 fatty acids. The absence of PA and PI is 289 unsurprising as these lipid classes are not common in algae (Guschina and Harwood, 2009) and not 290 generally measured in environmental samples (Brandsma et al., 2012; Schubotz et al., 2009; Van 291 Mooy and Fredricks, 2010). As all three community samples were taken during the summer 292 phytoplankton bloom, the relative abundances of the lipid classes did not differ significantly. 293 However, temporal shifts were observed in the relative abundances of individual lipid species within 294 each class. Such differences in fatty acid compositions are linked to either changes in the microbial 295 community structure, or adaptations to changing environmental conditions (e.g., temperature, light, 296 nutrients).

297

### 298 4 Conclusions and outlook

Although not an exhaustive analysis of marine or phytoplankton lipidomics, this study represents the first application of UHPSFC for the separation of all major phytoplankton (and plant) lipids, and clearly demonstrate the analytical power of the UHPSFC-ESI-MS/MS platform. The main advantages compared to other lipid separation methods are the improved chromatographic resolution and analysis speed of UHPSFC. This enables much higher sample throughput and thereby larger-scale and/or in-depth lipidomics studies. In addition, a wide range of column chemistries are available and it is possible to measure both polar and apolar lipid classes in a single analysis. Finally, the reduced

10

- analysis time and solvent use makes the system cheaper and more environmentally friendly to run. 307 Due to the comparative novelty of the technique, many of its aspects and applications are still being 308 explored. For example, alternative stationary phases may give better chromatographic peak shapes 309 for the galactolipids, different solvent combinations and gradients may yield better separation of 310 specific lipid classes, and the effects of temperature and pressure gradients on the chromatography should be investigated further. Finally, UHPSFC is likely to provide similar benefits to other organic 311 312 compounds, such as cellular metabolites, complex organic matter mixtures (soil and aquatic 313 P/DOM), or lipid biomarker (palaeoenvironmental) proxies. As a result of the advantages outlined 314 herein, UHPSFC will see increasing use instead or alongside of other chromatography methods in a
- wide range of research areas.

#### 317 Acknowledgements

- We thank Prof. Benjamin van Mooy for providing the DGCC standard and the Rothera marine
- assistants for their help with obtaining the RaTS phytoplankton community samples. Phytoplankton
- culture samples were kindly provided by colleagues at the University of Southampton: Dr Chris
- Daniels (*E. huxleyi*), Dr Andreas Johansson (*D. tertiolecta*) and Ms Despo Polyviou (*Synechocystis sp.*
- and T. erythraeum).
- This work was supported by the Engineering and Physical Sciences Research Council (EPSRC) Core
- Capability for Chemistry Research (EP/K039466/1), as well as the Netherlands Organisation for
- Scientific Research (NWO) Netherlands Antarctic Research Programme (851.20.047) and Netherlands
- *Polar Programme* (866.12.404).

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
