# Peer review of "Technical Note: Rapid Normal-phase Separation of Phytoplankton"

_Biogeosciences, 2016_

## Referee Comment (RC1) · H.F. Fredricks (Referee) · 18 Feb 2016

Review of bg-2016-13 Submitted on 12 Jan 2016 Technical Note: Rapid Normal-phase Separation of Phytoplankton Lipids by Ultra-High Performance Supercritical Fluid Chromatography (UHPSFC) J. Brandsma, T. R. Sutton, J. M. Herniman, J. E. Hunter, T. E. G. Biggs, C. Evans, C. P. D. Brussaard, A. D. Postle, T. J. Jenkins, and G. J. Langley

The authors present a technical note wherein they attempt to characterize the lipids of marine phytoplankton, along with phytoplankton community samples collected in Antarctic waters during a phytoplankton bloom. The novelty here is that that the method employs ultra-high performance supercritical fluid chromatography (UHPSFC) coupled to a triple quadrupole mass spec. This underutilized branch of chromatography affords

a high resolution normal-phase type separation with highly specific detection of lipids of the triple quadrupole mass spec. My overall impression is that the method has great potential, but as presented here there are sufficient major deficiencies that render the method unfit for purpose as it presented, that is, a method for analyzing the lipids of phytoplankton, and therefore I must recommend rejecting the manuscript.

In detail; general comments: The authors give no quantitative response factors or limits of detection for any of the lipids observed in spite of authentic standards being obtained which would allow them to easily do this. This in itself is not an insurmountable problem, the authors could easily quantify standards and add details to this note, but this knowledge would perhaps have alerted the authors to the bigger problem that is the crux of my review: the inadequate response of the glycolipids MGDG, DGDG and SQDG, rendering them unobserved in several phytoplankton cultures! Phytoplankton are by definition organisms capable of photosynthesis. Table 2 describes two phytoplankton as no MGDG or SQDG observed, and all others except the synechocystis as having relatively low amounts. The photosynthetic membranes are well known to contain MGDG,DGDG,SQDG and PG (Wada and Murata 1998) and there are numerous references in the literature that quantify the lipids of these organisms in detail (e.g. Abidi et al, Plant Physiology 2015: Van Mooy and Fredricks, GCA 2010) and show absolutely, that MGDG and SQDG the most abundant lipids in photosynthetic membranes. Since glycolipids are the bulk components of photosynthetic membranes a method published to examine phytoplankton surely must be able to sufficiently detect glycolipids?? Without any quantitative data it is impossible to know for sure, but there are several clues to suggest that the glycolipid response is very low, apart from not observing it in several well-characterized cultured organisms: the noisy baseline in the chromatograms and the peak shape is very poor – I'm not convinced it is due to poor chromatography as it told in the text– in my experience normal phase chromatography of glycolipids gives good peak shapes. I can think of several possible causes / remedies for this; the glycolipids are analyzed as their ammonium adducts, with the high flow-rate and large volume of make-up solvent containing a surprising 1% formic acid (more commonly

0.1% formic or 1% acetic), perhaps ammonium adducts are not the major ions of the glycolipids? Sodium adducts can be a problem even when ammonium is present in optimum conditions, did the authors look at the speciation of analyte/adduct ions? Neutral loss scans on a triple quadrupole mass spec are the least sensitive type of scan since both Q1 and Q3 are required to scan in tandem, and so sufficient scan time (0.5 seconds on our Thermo TSQ) is essential to obtain satisfactory sensitivity – the authors do not present sufficient analytical detail to comment fully on this, was this optimized? Presumably so, since the response PE and PG are more than satisfactory. I have no personal experience of SFC but it seems like the flow rates are very high for an electrospray method – typically flow rates are reduced to 'concentrate' the analyte yet 0.45 mL/min is added post-column, perhaps reducing this would yield further sensitivity?

Specific detailed comments:

Line # 121: Institution, not Institute.

Using both acetate (ammonium acetate in co-solvent) and formate (formic acid in the make-up) would lead to a confusion of anion-adducts. . . presumably PA is observed as both formate and acetate?

Capital L for litres/liters is easier to read than l I think.

Table 2 has very little meaning. Presumably this is based on peak area or peak height? Even a relative comparison of such data where the data is present from positive and negative ion mode and parent scan and neutral loss scan is impossible with no quantitative element! Of course PC is the most abundant – PC gives a very strong response since the 184 ion is often 1:1 abundance compared to the precursor ion! Consequently, the view of the membrane lipid profile is completely and unacceptably skewed by the lack of any consideration of quantitative response. Popendorf et al, (Lipids 2013), is a useful reference here, for example, MGDG is 2 orders of magnitude less sensitive that PC, under the given conditions.

The chromatograms in figure 1 would be better presented with some indication of absolute abundance. It would be useful to see chromatograms of the cultures and environmental samples too.

Section 2.3: No details are given for the MS settings; gas flows, temperatures, etc which would be useful to reproduce this method.

Conclusion: I suspect that further method development would increase the sensitivity of the glyco- and other lipids to a sufficient degree. However, given the method development that is required, and consequent re-analysis of standards and samples I feel that "major revisions" does not adequately cover the necessary complete re-do of the work and so must sadly recommend rejection of this note.

I have spent a great deal of time considering my response. I hope that by not making this review anonymous the authors will know and appreciate my intimate knowledge of phytoplankton lipids. I also hope my comments prove useful to the authors and I would be happy to correspond privately over the fine details. I look forward to seeing a sensitive, quantitative SFC method in the future and would be pleased to review such a manuscript.

---

## Referee Comment (RC2) · Anonymous Referee #2 · 22 Feb 2016

Review of bg-2016-13 submitted on 12 Jan 2016 Tecnical Note: Rapid Normal-phase Separation of Phytoplankton Lipids by Ultra-High Performance Supercritical Fluid Chromatography (UHPSFC) J. Brandsma, T. R. Sutton, J. M. Herniman, J. E. Hunter, T. E. G. Biggs, C. Evans, C. P. D. Brussaard, A. D. Postle, T. J. Jenkins and G. J. Langley

In this technical note the authors presented the results of the characterization of lipids in the cultures of marine phytoplankton and in the community of phytoplankton sampled in Antarctic waters during the time of austral summer by Ultra-High Performance Supercritical Fluid Chromatography coupled to a tandem quadrupole MS. Although it seems that the method should work, the characteristic distribution of phytoplankton lipid classes were not obtained (except for Synechocystis sp.). Since the reasons concerning the method is thoroughly and expertly commented by H.F. Fredricks I would add another possible reason for not finding the typical distribution of lipids in phytoplankton samples.

The lipid distribution shown in Table 2 (although it is very difficult to draw conclusions on the basis of +/- presentation) is more characteristic for the bacteria and/or detritus than for eukaryotes. So, it is possible that the most of the phytoplankton cultures decayed and dominated by fast-growing bacterial population. I wonder how the authors checked the phase of phytoplankton growth; did they count the cells under microscope and make sure they are growing the species obtained or conclude about the phase according to culture turbidity? Also in the samples from the Antarctic waters, (it seems that the quantity of filtered water is too small if there was no bloom), do they know the phytoplankton abundance or community composition? Here, it is also possible that phytoplankton was not dominating.

Although the manuscript has a lot of shortcomings and it is not now acceptable for publication, I would encourage the authors to repeat the experiments with cultures and try to find out the real reasons for such results.

---

## Author Comment (AC1) · 13 Apr 2016

Dear Dr Fredricks and Anonymous Referee,

Thank you for reviewing our manuscript on the separation of phytoplankton lipids by ultra-high performance supercritical fluid chromatography (UHPSFC). The aim of this paper is to introduce a new, and in our opinion very powerful and versatile, separation technology to the readers of Biogeosciences. It is not intended as "an attempt to characterize the lipids of marine phytoplankton", which is something we expressly stated in the main text (lines 235-237, line 299). As the manuscript title indicates, our focus is thus on the novel chromatographic aspects, and not on the subsequent detection or quantification of the lipids per se. For this there are a multitude of methods, instruments and bioinformatics approaches available, and we routinely use state-of-the-art mass spectrometry to measure a wide variety of lipid classes at the lowest possible levels. However, we do not contend the referees' key point that the results of the phytoplankton screening are inconsistent and at times very poor. The reasons for this do not lie with the UHPSFC separation, but rather with the MS detection. All method development was done on a single UPC2-TQD system, specifically obtained to trial the use of this novel type of chromatography for a wide range of analytes. This MS was not purposefully set up and optimised for lipid analysis, nor does this specific build of detector have the best levels of sensitivity. The fact that the results did not necessarily reflect the lipidomes of the screened phytoplankton species, which are indeed well-established and familiar to us, was therefore not very surprising. As indicated, the purpose of measuring a limited number of phytoplankton extracts was to test the use of UHPSFC chromatography on real-life samples. Optimising the MS detection and establishing a sensitive and (as far as this is possible) quantitative method was unfortunately not possible at the time, although we do recognise the need for this. To this end, we have now installed a dedicated UHPSFC system for use with our highest-end MS systems. We are in the process of replicating the method presented in this paper, this time with a fully validated MS method, and will be in a position to present these results in the very near future. Based on the editorial decision we are happy for this to be in the form of an amended manuscript or a full (re)submission. Finally, we thank the referees for highlighting this specific issue with our results and thereby strengthening the resubmission of our work.

Specific comments:

* With regards to the galactolipid chromatography we want to point out that the peak shape, while not as sharp as for example the phospho- or betaine lipids, is still at least as good as that obtained in regular LC systems. Figure 1 may be somewhat misleading as the galactolipid (and betaine lipid) peaks presented there are from a complex mixture of different lipid species (lines 493-495), whereas those in the upper two panels

are SRM traces of individual phospholipid species. This figure will be updated to show the chromatography of both individual lipids and whole classes in natural samples.

* Although we did not observe any deleterious effect from the use of formic acid in the make-up solvent, we have now modified this to ammonium acetate to be on the safe side.

* The specified high flow rates are typical for UHPSFC and no negative impact on electrospray ionisation efficiency has been reported to date.

* Line 121: changed

* Full MS settings will be added to the revised manuscript.

* Finally, with regards to accurate quantification of phytoplankton lipids in LC-MS methods, we want to point out that this is possible for only a very small number of lipid species for which isotopically labelled reference standards are available. Methods that work around this problem are certainly feasible (i.e. Popendorf et al. 2013 Lipids; Brandsma et al. 2012 Biogeosciences), but these do not fully account for effects such as species/class-specific differences in ionisation efficiency or concentration-dependent ion suppression in the chromatography. It is our hope that more synthetic standards will become commercially available to support this type of work.

Yours sincerely,

Joost Brandsma

―――――――――――――――――――――――――――